# 'Children awaken by playing': a qualitative exploration of caregivers' norms, beliefs and practices related to young children's learning and early childhood development in rural Burkina Faso

Mari Dumbaugh,[1,2] Mireille Belem,[3] Sylvain Kousse,[3] Patricia Ouoba,[3] Adama Sankoudouma,[4] Achille Mignondo Tchibozo,[4] Pasco Fearon,[5,6] Jennifer Hollowell ![ORCID],[7] Z Hill ![ORCID],[8] SUNRISE team

For numbered affiliations see end of article.

**Correspondence to**
Dr Z Hill; z.hill@ucl.ac.uk

## ABSTRACT

**Introduction** Evidence suggests that responsive caregiving and early learning activities positively impact developmental outcomes, with positive effects throughout the life course. Early childhood development interventions should align with local values, beliefs and resources but there has been little research of caregiver beliefs and perspectives on development and learning, especially in sub-Saharan Africa. This qualitative study explored norms, beliefs, practices and aspirations around child development of caregivers of young children in rural Burkina Faso.

**Methods** We conducted 32 in-depth interviews with mothers and fathers of young children and 24 focus group discussions with mothers, fathers and grandmothers, which included trying behaviours and reporting on experiences. The research informed the development of Scaling Up Nurturing Care, a Radio Intervention to Stimulate Early Childhood Development (SUNRISE), an early child development radio intervention.

**Results** Caregivers described a process of 'awakening', through which children become aware of themselves and the world around them. Perceptions of the timing of awakening varied, but the ability to learn was thought to increase as children became older and more awake. Consequently, talking and playing with babies and younger children were perceived to have little developmental impact. Caregivers said children's interactions with them, alongside God-given intelligence, was believed to impact later behaviour and development. Caregivers felt their role in helping their children achieve later in life was to pay for education, save money, provide advice and be good role models. Interaction and learning activities were not specifically mentioned. Caregivers who trialled interaction and learning activities reported positive experiences for themselves and their child, but interactions were often caregiver led and directive and play was often physical. Key barriers to carrying out the behaviours were poverty and a lack of time.

## STRENGTHS AND LIMITATIONS OF THIS STUDY

⇒ The use of different qualitative methods, including *Trials of Improved Practices,* allowed us to explore social norms and beliefs of early learning and caregiving as well as how norms and beliefs played out in caregivers' lived experiences.
⇒ Our South-North research team prioritised intentional space for iterative reflexivity and colearning, ensuring our research tools, training of data collectors, approaches to data collection and analysis were triangulated and validated from multiple positionalities.
⇒ Logistics prevented us from speaking to caregivers under the age of 18, so our results do not reflect perspectives from this important caregiver group in the Burkinabé context.
⇒ Security concerns limited our ability to incorporate participant observation into our study which could reveal more nuanced information relevant to early childhood intervention design.

**Conclusions** Exploring early childhood beliefs and practices can reveal important sociocultural beliefs which, if incorporated into programme planning and implementation, could help achieve more impactful, acceptable and equitable programmes.
**Trial registration number** NCT05335395.

## INTRODUCTION

Evidence suggests that responsive caregiving and early learning activities positively impact early childhood development (ECD) outcomes, educational achievement and economic opportunities throughout the life course.[1–6] Supporting caregivers to provide responsive care and learning activities during children's early years is considered a key strategy for

achieving equitable health and development through platforms such as individualised home visits, group interventions and primary healthcare contacts.[1 5 7]

Parental ethnotheories, or 'the shared beliefs about the goals of child development and the socialisation practices that will achieve these goals' (Greenfield and Keller,[8] p7) can vary greatly between settings.[3 8–15] Despite progress in developing ECD measures relevant across cultures,[16] much of developmental science that informs intervention design originates from research in high-income, Euro-American countries by researchers and with children of the same demographic.[10 13 17–19] There is general consensus that ECD interventions require adaptations for different sociocultural contexts to ensure the development of ethical and effective interventions.[13 17 19–24] A 'one size fits all' approach to ECD curricula and interventions risks missing or even damaging important context-specific beliefs, cultural values and social cohesion.[10 13 17] Debates continue, however, as to how to counter research biases and what they mean for programming and measurement.[13 17 18 20 25–27]

Formative research has informed the design of ECD interventions in low and middle income countries.[12 15 21 22 28–34] However, few trials in sub-Saharan Africa reported using formative research or local community advisory boards in intervention design, and there has been little research exploring caregiver beliefs and perspectives on development and learning from birth.[12 15 22 34–38] We conducted formative qualitative research in rural Burkina Faso to explore ECD norms, beliefs and practices and aspirations of caregivers of children aged 0–3 years. This research informed the development of *Scaling Up Nurturing Care, a Radio Intervention to Stimulate Early Childhood Development* (SUNRISE), running from 2022 to 2025 and offering caregivers ECD information via radio broadcast to encourage responsive engagement with young children (Trial registration: Clinicaltrials.gov identifier NCT05335395).

## METHODS

We conducted semistructured, in-depth interviews (IDIs) with mothers and fathers and focus group discussions (FGDs) with mothers, fathers and grandmothers. FGDs allowed us to explore social norms and beliefs and IDIs to understand how these played out in caregivers' lived experiences. Data were collected in four villages in Balé and Boulgou provinces, with equal numbers of participants sampled from each village (see table 1). Villages were safe for research teams and did not have exceptional characteristics such as being located along a major thoroughfare. They had a range of ethnicities to complement previous data we had collected on the country's dominant ethnic group (the Mossi).[15]

### Ethics

The Burkina Faso Institute for Health Science Research Ethics Committee and the University College London Research Ethics Committee granted ethical approval for this study.

### Sampling and data collection

Data were collected from March to April 2021. Data collectors and supervisors (women and men) had completed 2–4 years of undergraduate study and had previous experience collecting qualitative data. They underwent 3 weeks of training to review methods, ethics, research tools and transcription. We conducted a pilot study in Ouagadougou to finalise the data collection guides. Supervisors were responsible for logistics, recruitment and data quality; data collectors obtained informed consent and conducted IDIs and FGDs.

Supervisors used the same procedure to recruit for IDIs and FGDs. First they introduced the study to village leaders who identified a community informant knowledgeable about the community, but not a member of the village chief's family. The informant, often a community health worker, then accompanied the supervisor to households with eligible children, where supervisors introduced the study and provided an information sheet detailing the study objectives, confidentiality, participants' rights and research team contact numbers. For participants unable to read French, information and consent forms were read aloud with a non-research team member witnessing and cosigning. For those unable to write, a thumb print replaced a signature. Supervisors purposively selected

| Table 1 | Participant sociodemographics | | | | | | |
|---|---|---|---|---|---|---|---|
| **Research activity** | **Participant group** | **Total number of participants** | **Participant age range** | **Range of number of children\*** | **Age ranges of children\*** | | |
| | | | | | **0–11 Months** | **12–23 Months** | **24+ Months** |
| In-depth interviews | Mothers | 16 | 20–40 | 1–7 | 6 | 8 | 2 |
| | Fathers | 16 | 23–59 | 1–15 | 8 | 7 | 1 |
| FGDs† | Mothers | 28 | 18–36 | 1–6 | 13 | 8 | 0 |
| | Fathers | 26 | 24–58 | 1–10 | 17 | 9 | 0 |
| | Grandmothers | 27 | 40–68 | 1–10 | 20 | 7 | 0 |

\*For grandmothers, refers to number/ages of grandchildren.
†Five to seven participants per FGD.
FGDs, focus group discussions.

participants over 18 years old, ensuring that a range of children's ages were represented: 0–3 years for IDIs and under 24 months for FGDs as previous research shows caregivers are less likely to engage interactively with this age group in Burkina Faso.[15 29]

IDIs were conducted in a private place in participants' homes and explored typical daily routines, parents' perceptions of children's early learning and parents' short-term and long-term aspirations for their children (see online supplemental appendix 1). Participants were given soap after each IDI to thank them for their time. FGDs were stratified by participant group to allow topics to be discussed from a perspective of similar positionalities. Two FGDs were conducted, in a neutral space away from other community members, with each participant group in each village by a facilitator and note taker.

FGDs used an adapted *Trials of Improved Practices* (TIPs) methodology.[15 39] A first round explored norms and beliefs around early child learning and caregiving (see online supplemental appendix 1). Then, participants listened to four recordings on the impact of early experiences on development, the importance of talking to children from birth, play as a vehicle for early learning and development and the benefits of frequent praise, encouragement and affection (see online supplemental appendix 2). Participants shared their reactions to the spots and were invited to practise talking, playing or praising/encouraging their child at home, and return for a second FGD in 1 week's time to share their experiences, challenges and child, family and community member reactions to the behaviours. Participants received money for transport. Four participants were lost to follow-up in the second FGDs (two fathers, one mother and one grandmother).

Research activities lasted 45–90 min, were conducted in local languages (Bissa or Dioula) and audiorecorded. Given local gender norms which might have induced response bias between researchers and participants of different genders, data collectors were paired with participants of their same gender.

Table 1 details participant sociodemographics (see also online supplemental appendix 3). Participants were married or widowed and identified their religion as Muslim, Christian or animist/traditional. Ten ethnic groups were represented in the sample population, with about half of participants identifying as Bissa. Participants' education levels ranged from none to high school completion. Most educated participants had completed some or all of primary school. The vast majority were farmers.

### Data transcription and analysis
Audio data were translated and transcribed into French. To ensure data quality, the first interviews were double transcribed by data collectors in the field and Ouagadougou-based transcriptionists. Senior researchers compared transcriptions for accuracy, giving translators feedback on translations and interviewers feedback to enhance their reflexivity and improve probing. Throughout data collection audio and transcripts were reviewed to ensure accurate translations.

Transcripts were analysed by francophone (MB, MD, SK, PO) and anglophone (ZH, MD) senior researchers. The non-francophone researcher (ZH) translated transcripts using Google Translate. We used a deductive–inductive approach to content analysis. First, team members read three IDI and three FGD transcripts to ensure familiarisation with data and themes. A bilingual researcher deductively developed initial code books for each data collection method based on the topics covered in the research guides. Then, a selection of IDIs and FGDs were double coded inductively within these broad themes to develop subthemes and facilitate standardised coding between team members. The team discussed any variation in coding and resolved differences via consensus discussion. Once coding was standardised and adequately detailed, the remaining transcripts were divided between researchers for analysis. We held frequent meetings to discuss findings, clarify translations, iteratively refine the code books and confirm that saturation was reached within our planned sample size (additional data collection would not have produced new information). Francophone researchers used NVivo (released March 2020) to complete analysis; the anglophone researcher coded by hand.

### Participant and public involvement
Participants and/or the public were not involved in the design, conduct, reporting or dissemination plans of this research. Final transcriptions and findings were not sent to participants, however participants were given contact numbers of research team leads should they want to review data or results.

## RESULTS
Analysis resulted in four major themes: perceptions of young children's consciousness; when and how children become conscious and learn; impacts of caregiver–child interactions; and parents' aspirations for their children. We have noted where differences in results were observed by caregiver role. Caregiver perspectives reported in this paper did not differ by the caregiver's or child's gender (child's gender only collected for IDIs), or between ethnic groups.

### Perceptions of young children's consciousness
Caregivers consistently conceptualised the development of child consciousness as a process of 'awakening': a child becoming aware of themselves and the world around them, their intellectual development, increasing capacity to understand and act with intelligence, their ability to follow directions, to understand the difference between right and wrong and to act on sociocultural norms such as greeting people, sharing food with others and bringing visitors water.

When a visitor arrives at the house, the child quickly goes to greet the visitor and gets water for him…The child tries to do what [adults] do around him. In doing so, he…[shows he] is truly awake. (Grandmother; grandchild 4 months)

The process, timing and speed of awakening was unique to each child and influenced by the 'God-given' nature of the child, caregivers' behaviours, or a combination of both.

### When and how children become conscious and learn

A major finding was caregivers' differing perceptions as to when children began to awaken and learn, varying from the earliest months of life to 5 years of age and the start of formal schooling. Caregivers said babies were not awake from birth as they were born without knowledge, memory or skills. The few caregivers who said that babies were capable of learning new things in their first months cited various signs of early learning: smiling and laughing; 'playing' with their hands; crying for a diaper change; following people with their eyes; recognising people; and acquiring new motor skills.

Perceptions of the timing of children's language comprehension also varied, from the earliest months of life to older than 2 years. Most caregivers thought that children usually start to comprehend language after about 1 year old, and show it by repeating and remembering words:

At 15 months the child begins to grasp some words and understand them. First he mixes up words, he calls people but…you cannot understand who he is calling. At 16 or 17 months he begins to talk more clearly, he knows how to call his father and mother, and that progresses gradually until he is 20 or 24 months old. (Father; child 33 months)

Fewer caregivers thought that preverbal children could learn to recognise and engage in forms of communication such as understanding their mother's expressions and tones, recognising when people were talking or being quiet and comprehending simple instructions from adults.

A child knows words even when he is small. Even at five months, if you say something, [the child] hears but he cannot speak…Sometimes when you say certain things,…[the child] starts to cry. He cries because he understood something. [The child] begins…to hear words without you realising it. (Father; child 11 months)

Caregivers said children learn to talk through interactions such as call and response, imitation, repetition and play with adults and other children.

Talking to young babies in a conversant style was not common because of the perception that babies did not yet understand words and could not respond. Caregivers felt that talking to a preverbal baby was like talking to yourself. FGD participants highlighted this challenge after trying TIPs behaviours in their homes:

What was difficult for me was when the child said "ha ha ha!" That means that he wants to talk to us, but we do not understand what he says, so we tell him to stop…and be quiet. As a Bissa saying goes, "A mute person cannot say to his grandmother, 'I want such and such a thing.'" (Grandmother; grandchild 1 month)

Caregivers also said some community members felt that talking to preverbal babies or playing with children was bizarre or a waste of time, with no real benefit to the young child.

I was sitting with my child in the middle of the day and I heard a knock at the door. The person [at the door] asked me who I was talking to, and I said I was talking to my baby…I told her my baby enjoys it when I talk to her…The person replied, "What does this little child know for you to speak [to her]?" I told her [my baby] understands and that's why I'm chatting with her. [My baby] laughs and she appreciates it, and I like it too. The person…[told] me that the [only] reason I am [talking to my baby] is that I don't have a job. (Mother; child 1 month)

Despite divergence on when exactly awakening and language comprehension begin, a major finding was that caregivers agreed on the learning process itself. They said young children learn 'little by little', learning more and more quickly as they awaken and are older. Children learnt through observation, recall, imitation and repetition of others; talk and explanations by caregivers; play on their own or with others; being corrected or encouraged; and at school through formal teaching. Talking to verbal children was believed to help children understand and learn words and recognise loved ones.

A child who is not used to having people talk to him will be less intelligent. But, a child who has always had people talking to him, that is like school for him and he will do remarkable things. You will see he is truly awakened. (Grandmother; grandchild 1 month)

Play, or 'children's work', alone or with peers, was considered particularly important for development and learning as it helped awaken children and 'open their spirit', helped children learn new words, expanded motor skills such as sitting up, clapping hands, crawling, walking and dancing; encouraged independence; made children intelligent by providing new experiences, socialisation with others and taught children new ideas and lessons such as dangerous things to avoid (ie, fire); and was an opportunity to practise life tasks through pretend games like cooking, playing shop, washing clothes and farming.

Play awakens children. If a child plays, he learns many things…Play also develops his intellectual capacities. (Mother; child 2 months)

Children usually played independently or with siblings and peers. Play between adults and children was mostly physical such as bouncing up and down and tickling. Other forms of play were not widespread, but caregivers who played with their children after FGDs had positive experiences.

Awakening and learning were at once influenceable and predestined. For example, while caregivers said play and talk could contribute to children's early or quick awakening, they also said these interactions would have limited influence until children were sufficiently awake (usually around 2–3 years old). Most caregivers also said 'God given' or innate intelligence, character traits or circumstances determined children's development and learning.

> God plots the future of each child. (Grandmother; grandchild 4 months)

> Some kids get better at a game the more they play. There are other children who sit and do not know how to play; everything is in the hands of God. (Mother; child 21 months)

A child who 'lacks blessings', for example, may be a difficult child or slow to follow instructions. Or, the quality of a woman's breast milk could positively or negatively impact her children's developmental achievements.

While recognising that each child's developmental path was unique, caregivers had expectations around the timing of developmental milestones, and noticeable delays were cause for concern:

> Children…do not all have the same growth rhythm. At six months, some are sitting up and crawling… There are other [children] who do not grow quickly. At 1 year they do not want to walk and it scares you… (Mother; child 12 months)

### Future impacts of caregiver–child interactions

Caregivers believed early experiences could impact a child's later behaviours and development, and that positive caregiver–child interactions were beneficial in the short and long terms.

> Some women insult their children…, for example, 'Idiot, God will punish you!' But, the child hears what you say without being able to respond, and he grows up with the words spoken to him. So, if we speak to the child using [kind] words, he will grow up with that. (Mother; child 18 months)

The perceived effects of disciplinary practices demonstrated the same. While caregivers universally agreed that sanctions and discipline were important aspects of child rearing, many also said that too frequent or too harsh discipline could have negative consequences for the child and their relationship with their parents well into adulthood.

The positive, long-term impacts of speaking to children softly, with positive words and encouragement were emphasised:

> When you scold a child a lot, he gets confused; so you have to speak to him softly…this is when his intelligence will increase. (Mother; child under 6 months)

FGD participants said children reacted positively to talk, play and praise during the TIPs. Even very young babies reacted to ECD behaviours through direct eye contact, giggling, babbling and moving their arms and legs:

> When he sees me he looks at me with big eyes and he moves [to show] he understands. When he sees me, he jumps, he laughs and he looks at me; so I know he understands. (Mother; child 5 months)

Older children's reactions to ECD behaviours tended to be more pronounced. After engaging in ECD behaviours, caregivers said older children responded more positively to them, were more aware of their presence, ran to greet them and asked them to play again. They said children were happier, more awake, had learnt to play or played better than before, increased their understanding and level of communication, had improved attitudes, followed directions, cried less and were more likely to interact and share with others.

> Before [I tried the ECD practices], …my child…did not want to share with anyone else. But, now, with the little time that I have played with him, he has gotten used to me…His mother bought him peanuts and…I asked him to give his grandmother one and he did. This is how I know he recognizes me now, as his father, thanks to the time I spent with him. (Father; child 30 months)

While these tangible changes were positive, the interactions caregivers described tended to be caregiver-led and directive in nature, rather than child-led and responsive to the child's signs and reactions. For example, talking to the child to give directions or monologuing, rather than engaging in a conversational exchange with the child; exclusively physical play (ie, lifting child in air, tickling); or, giving children objects to keep them busy or console them rather than facilitating play with the child.

### Parents' aspirations for their children

IDI participants were asked to describe their aspirations for their children. In the short-term, parents wanted happy, healthy children who were successful at school. They wanted their children to be close to them, respectful of elders and visitors, helpful without being told what to do and good at following instructions and advice. They also described desirable character traits: independence, curiosity, politeness, active and alert (especially at school) and playful, as play signalled a healthy and happy child.

In the longer term, parents wished for their grown children to be healthy, continue practising their religion, marry a good person and have children themselves.

Parents wanted children to be 'better off' than they were financially by acquiring money, land, a house and animals, and hoped their children would visit them often and be able to support them financially in old age. Many parents wanted their children to learn a trade, or to have a particular profession such as a government worker, health provider, entrepreneur, school teacher, religious teacher, soldier or police.

> If you put [your child] in school, he can be awakened and study, go farther than [you]. (Father; child 16 months)

Parents also hoped their children would become respected and forgiving community members, as their success as parents would be measured by their adult children's esteem in the community.

> I would like her to be forgiving…care for others. If you do not have a good heart and you do not care for others, no one will become close with you. (Father; child 14 months)

Parents said they could help facilitate their children's success by paying for their education, saving money for them, guiding them with good, moral advice and being good role models. But, they rarely mention ECD behaviours as something they could do to facilitate success:

> It is up to us [parents] to show them, from a young age, what to do and what not to do to succeed in their life. All of this is through advice, if you do not guide your child, he won't succeed in life, and help you [in your old age]. (Mother; child 10 months)

Childrearing in extreme poverty presented parents with incessant challenges. A lack of food, treating household illnesses and injuries, having a baby before the youngest child had grown and maintaining patience with children also added to household difficulties.

> Raising children, it's not easy - go to the hospital, eat, it's all very complicated for us. They say that [health services] are free, but they don't give you [free] medicines, so you are there suffering with your child. If you do not have money, it is exhausting to be a parent. (Father; child 8 months)

Parents said the constant need to seek work, money and provide for the household left them with little time to practise ECD behaviours with their young children as often as they would like. Women in particular mentioned the effects of their high burden of housework, and some men worked away from home for extended periods of time.

> My work is too much. During the day I have to prepare [food]; to pray; washing the children. Women [in our community] have a lot of work. (Mother; child 12 months)

## DISCUSSION

This study expands the evidence base documenting ECD norms, beliefs and practices for young children in sub-Saharan Africa, while also contributing to calls for culturally relevant adaptations of ECD curricula and programming for non-Western contexts.[9 10 13 17 19 40] Evidence demonstrates the positive effects of a standard set of ECD behaviours across different cultural contexts,[1 4 27 41] but authors argue that 'imported' ECD curricula 'cannot fully meet the needs of African families' (Prochner and Kabiru, p130)[42] as they will not reflect fundamental cultural and conceptual differences between Western parenting beliefs and practises and the diverse parental ethnotheories across African contexts.[13 14 17]

The SUNRISE intervention design team, a mix of local and UK-based mass media communication experts, used our findings to highlight the benefits of ECD behaviours in alignment with local values, beliefs, practises and aspirations in a rural, sub-Saharan Africa context. Following the *Saturation+* approach to behaviour change, these briefs were scripted into dramatic sketches, then locally produced and broadcast via radio in seven Burkinabé languages multiple times a day.[43] Local radio presenters were also trained to host live, call-in talk shows on ECD-related topics.[44]

SUNRISE messages encourage responsive interactions with children from birth by integrating and validating local normative perceptions and beliefs, using terms and scenarios that are salient to the population in rural Burkina Faso. For example, caregivers' belief that ECD behaviours would have a positive impact on their children's development in the short-term and long-term became a central SUNRISE message. The concept of 'awakening,' which echoes descriptions of child development in other similar settings,[14] defined most discussions about ECD in our study. Therefore, SUNRISE broadcasts use local perceptions of 'awakening' to explain the principles of ECD and encourage responsive caregiving. Caregivers looked positively on children who awakened early and quickly, so messages emphasise the positive effects of caregivers' responsive engagement with children on the timing and speed of awakening. Formative research revealed caregiver–child interactions were directive in nature, so radio messages encourage child-led play from birth. Radio messages also integrate local beliefs which might hinder the practice of ECD behaviours. Given the poverty-induced time constraints many families faced, broadcasts encourage play during daily tasks such as bathing. Messages also respond to the perceived limited impact of responsive interactions on babies in the first months of life, a belief also reported elsewhere in Burkina Faso and Senegal,[12 29] and that it is 'crazy' or a waste of time for adults to play with babies.

Parents' aspirations for their children's development and future success also aligned with documented benefits of responsive ECD interactions. Caregivers said they wanted their children to be articulate and confident in front of strangers and within the community, and

success at school was a high priority. After practicing ECD behaviours at home, caregivers said talking to and playing with children from a young age would help them 'have the right words', be successful at school and, later in life, be involved and respected community members. While we agree with authors who critique assumptions that populations across all sociocultural contexts share the same educational and aspirational goals,[10 13 45 46] our research affirmed that many parents in this context aspire for their children to have academic success leading to professional careers. SUNRISE messages, therefore, highlight success at school as one of many benefits of ECD behaviours.

The above examples demonstrate the importance of formative research which explores parental ethnotheories in non-Western contexts and if and how 'standard' ECD behaviours fit into context-specific parental and communal beliefs, practices and aspirations for children. Findings from this process should be legitimised in their own right, not necessarily compared with Western ECD standards to deduce what is 'lacking' or what needs to be 'fixed',[14] and integrated into intervention development and implementation. This makes programme messages more relevant and familiar to the target audience, an aspect of ECD intervention development that is important ethically, and also optimises programme efficacy.[17 23 40 47 48] Methodologically, our South-North research and implementation teams prioritised intentional space for iterative reflexivity and colearning, which greatly aided our overall adaptation process. We have also continued to collect qualitative data throughout the campaign, reflecting the need for and benefits of ongoing contextually sensitive adaptation and improvement.[49]

Ethnographic research such as participant observation[13] could reveal more nuanced information relevant to ECD which existing programming often does not discover or ignores,[17 45 50] but security concerns meant that this was not possible as part of this study. Logistical challenges also prevented us from speaking to individuals younger than 18 years old, an important limitation given high rates of adolescent pregnancy in Burkina Faso.[51] A limitation of the TIPs methodology in particular is that participants can easily identify 'desirable' behaviours with the risk of social desirability bias related to current practices and suggested behaviours.[52] We countered this risk by emphasising our desire to learn about all participant experiences, normalising potential challenges and focused probing. In future, the participation of local populations at every stage of formative research and intervention codesign[53] could be key in realising the highest level of ethical, culturally adapted, effective ECD programming. Finally, our findings are particular to a specific research context and are not necessarily generalisable.

Understanding the early childhood norms, beliefs and practices of a local population can reveal important sociocultural differences between assumptions underlying different ECD curricula and the contexts in which they are implemented. Acknowledging and integrating these differences during programme planning and implementation could help achieve more equitable child development and, subsequently, social, health and economic outcomes globally.

**Author affiliations**
[1]Institute for Global Health, University College London, London, UK
[2]Insight Impact Consulting, Chicago, Illinois, USA
[3]Research, Develpment Media International, Ouagadougou, Burkina Faso
[4]Innovations for Poverty Action Francophone West Africa, Ouagadougou, Burkina Faso
[5]Department of Psychology, University of Cambridge, Cambridge, UK
[6]Centre for Family Research, Department of Psychology, University College London, London, UK
[7]Development Media International Associates CIC, London, UK
[8]Institute for Global Health, University College London Research Department of Epidemiology and Public Health, London, UK

**Acknowledgements** Sincere thanks to the caregivers who took time out of their days to participate in this study and local authorities and leaders for their collaboration in facilitating access to their communities. Without them, these important perspectives would not be available to the wider scientific community. We would also like to thank the data collectors for their dedication and quality contributions to this study under demanding field circumstances.

**Collaborators** The following are members of SUNRISE (Scaling Up Nurturing care, a Radio Intervention to Stimulate Early child development): Seeba Amenga-Etego (Ghana), Pare Balary Touba (Burkina Faso), Mireille Belem (Burkina Faso), Radha Chakraborty (UK), Abbie Clare (UK), Cecily Cocks (UK), Sabin Dandjinou (Burkina Faso), Mari Dumbaugh (USA), Pasco Fearon (UK), Roy Head (UK), Zelee Hill (UK), Jennifer Hollowell (UK), Bassirou Kagone (Burkina Faso), Betty Kirkwood (UK), Sylvain Kousse (Burkina Faso), Alexander Manu (UK), Achille Mignondo Tchibozo (Burkina Faso), Joanna Murray (UK), Guikierba Namoano (Burkina Faso), Patricia Ouoba (Burkina Faso), Tom Palmer (UK), Reetabrata Roy (India), Adama Sankoudouma (Burkina Faso), Jolene Skordis (UK), Achille Tchibozo (Burkina Faso).

**Contributors** PF, JH and ZH designed the formative research reported on in this paper. MD and ZH designed the data collection tools with input from MB, PF, JH, SK and PO. MD and ZH designed data collector training and MB, SK, PO, AS and AT facilitated the training. AS and AT supervised data collection and monitored the quality of data, data transcription and data translation. MB, MD, ZH, SK and PO analysed data. MD and ZH drafted and revised the paper, with significant input from MB, PF, JH, SK, PO, AS and AT. ZH is guarantor of the study. All members of SUNRISE have made significant intellectual contributions to the SUNRISE campaign and trial design and execution, of which the formative research reported on in this paper was a part.

**Funding** This research was funded by the Wellcome Trust (grant numbers 215492/Z/19/Z [LSHTM] 215492/A/19/Z [DMI]) under the Scaling Up Nurturing Care, a Radio Intervention to Stimulate Early Childhood Development (SUNRISE) project.

**Competing interests** None declared.

**Patient and public involvement** Patients and/or the public were not involved in the design, or conduct, or reporting or dissemination plans of this research.

**Patient consent for publication** Not applicable.

**Ethics approval** This study was approved by the Burkina Faso Institute for Health Science Research Ethics Committee (2021-02-031) and the University College London Research Ethics Committee (2982/004).

**Provenance and peer review** Not commissioned; externally peer reviewed.

**Data availability statement** Data are available upon reasonable request. De-identified transcripts of semi-structured, in-depth interviews and focus group discussions are available from the corresponding author (z.hill@ucl.ac.uk) upon reasonable request. Data collection tools are included as supplementary material to this publication.

of the translations (including but not limited to local regulations, clinical guidelines, terminology, drug names and drug dosages), and is not responsible for any error and/or omissions arising from translation and adaptation or otherwise.

**ORCID iDs**
Jennifer Hollowell http://orcid.org/0000-0002-4041-5732
Z Hill http://orcid.org/0000-0002-2614-8877

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
