## [Reviewer comments · BMJ Open]

ARTICLE DETAILS

TITLE (PROVISIONAL)	'Children awaken by playing': A qualitative exploration of caregivers' norms, beliefs and practices related to young children's learning and early childhood development in rural Burkina Faso
AUTHORS	Dumbaugh, Mari; Belem, Mireille; Kousse, Sylvain; Ouoba, Patricia; Sankoudouma, Adama; Tchibozo, Achille; Fearon, Pasco; Hollowell, Jennifer; Hill, Z

VERSION 1 – REVIEW

REVIEWER	Mabetha , Khuthala University of the Witwatersrand Johannesburg
REVIEW RETURNED	12-Jul-2023

GENERAL COMMENTS	This is a very interesting study that addresses a very important gap and can thus be deemed suitable for publication. Abstract Introduction The introduction of the abstract has been well written but there are two important pieces of information missing: (1) From the whole background provided on responsive caregiving and its positive impact on developmental outcomes, what was the gap that merited further research into this area? i.e., gap related to why you conducted the study? (2) Secondly, what is the aim of the study? Please make the aim of the study explicit and place it right at the end of the introduction section. Methods "We conducted qualitative research in rural Burkina Faso to explore norms, beliefs, practices and aspirations", lines 21-22: I see that the aim of the study was placed in the methods section. Please move it up right at the end of the introduction section. Also, what stood out to me is that the title indicates that the authors explored "perceptions" of young children's learning and childhood development, yet the aim was to explore "norms, beliefs, practices and aspirations". All these terms are different to "perception" which refers to the manner in which a phenomenon is understood or interpreted whereas "beliefs" refers to the acceptance that something exists or is true. Same applies with "perception" and "practices" and "perceptions" and "aspirations". The title is fine but the "perceptions" term needs to be replaced with a term that captures the aim of the study "norms, beliefs, practices and aspirations". Main text Introduction "aged three four years", line 3 – I think this was meant to be three to four years old. Please fix it.
---

	“(Clinicaltrials.gov identifier NCT05335395”, line 28: This referencing style (if it is one) is different to the numbering system used throughout. Please be consistent (that’s if this is a referencing style and not a trial registration number). If it is a trial registration number, please be explicit and indicate that it is a trial registration number. Methods “We conducted in-depth interviews (IDIs) with mothers and fathers and focus group discussions (FGDs)”, line 32- A few questions: (a) How many IDIs and FGDs were conducted? How many mothers and how many fathers? What was the maximum number of people in each FGD? Did each FGD constitute only of mothers, fathers and only of grandmothers OR was there a combination of both mothers, fathers, and grandmothers? What was the rationale for this combination or lack thereof? (b) Given that IDIs were conducted with mothers and fathers of the children AND FGDs were also conducted with the mothers and fathers, including grandmothers, what was the rationale for using both data collection methods? Please elaborate on this. (c) How was data saturation reached? In other words, did you get to nth number of in-depth interviews perhaps due to redundancy in information provided by the participants or was nth number the sample size selected for a specific reason? Same applies to the FGD discussions. “in four villages in Balé and Boulgou provinces, Burkina Faso”, lines 33-34: I would suggest that you construct a table showing how many people were sampled in each of these four villages for both data collection methods (also show how many were mothers, fathers, grandmothers) as a supplementary table to refer to. I see you have Table 1 showing the number of participants but I think you should also show how many were sampled in each of these of these four villages for both data collection methods. “The Burkina Faso Institute for Health Science Research Ethics Committee and the University College London Research Ethics Committee granted ethical approval for this study”, lines 42-43: Sub-heading missing. Please indicate what this refers to. Either “ethical considerations” or “ethics”. Data collection “No participants left the study once they agreed to participate” line 13-14- I am not sure what the authors mean here. Are they saying once participants agreed to participate, they did not have the option to opt out of the study (suggesting that participation was only voluntary when consenting) or does this mean that out of all the participants who agreed to participate, none of them opted out of the study? Please correct this so that there is no ambiguity or more so that it is clear that participants were not forced to complete the study. “Data collectors were paired with participants of their same gender.”, lines 18-19- What was the reason for this? Of importance, were the IDIs and FGDs conducted only in English or were there instances where participants were interviewed in their native language? Data transcription and analysis
--	---

	“deductive-inductive approach”, line 28- I understand the use of an inductive approach but I am not sure why a deductive approach was also employed since there was no mention of a predetermined framework that was used in this study which also contributed to the development of the IDIs and FGD topic guides. The authors only mentioned that this research informed the development of Scaling Up Nurturing Care, a Radio Intervention to Stimulate Early Childhood Development towards the end of the introduction but did not explain how it informed its development. Thus, a deductive approach is not appropriate in this context. Please briefly explain the steps that you undertook in the inductive analysis which ultimately led into the generation of the themes and sub-themes.
--	---

REVIEWER	Wuermli , Alice New York University
REVIEW RETURNED	16-Jul-2023

GENERAL COMMENTS	This paper presents the process and findings of a qualitative study to inform the design and curriculum of a radio-based ECD intervention aimed at encouraging early stimulation. I very much appreciate this contribution and I thank the authors for the opportunity to provide my 5 cents. The paper is well written and structured, and methodologically as far as I can tell sound (with some requests for additional specificity and clarifications). Below I suggest areas for improvement. The paper points out the cultural biases in the research on child development in the Majority World and proposes to address these in this study in order to inform a culturally adapted, appropriate version of a radio program to promote ECD. Overall, especially later on in the Discussion, the paper does a good job at demonstrating how this process informed the design of this intervention. Earlier on though there are a few instances where I would like to see a bit more critique or caution. How are “we” defining and measuring responsive/sensitive caregiving and stimulation? How are “we” measuring child outcomes? Are these concepts and measures universally applicable and unbiased? And are we seeing impacts in interventions encouraging “stimulation” because stimulation always leads to better outcomes, or because our outcome measures are picking up on what we are “teaching” in these interventions? Just because an intervention targeting a certain behavior leads to better outcomes according to our measures doesn’t mean that the relationship holds universally, or that the research is culturally appropriate and valid. I would like to see the authors be a bit more critical in how they conceptualize development, developmental processes, and developmental outcomes and couch existing evidence within this framework. I get the sense that the authors understand and appreciate the cultural biases in ECD research, and I’d like to see this reflected throughout the paper. Eg., if outcomes are assessed with a Bayley, what does that mean for our ability to claim better child outcomes? Is the Bayley universally predictive of later outcomes? Who decided that the tasks on the Bayley are what matters, at a given age and stage of development? Furthermore, the Lu et al. paper uses DHS and MICS data which despite their tremendous contribution have not gone without criticism of being somewhat culturally insensitive. There is an extensive literature on culture and ECD that should be reflected (see for instance Scheidecker, G., Chaudhary, N., Oppong, S., Röttger-Rössler, B., & Keller, H.
---

(2022). Different is not deficient: respecting diversity in early childhood development. *The Lancet Child & Adolescent Health*, 6(12), e24-e25. [https://doi.org/10.1016/s2352-4642\(22\)00277-2](https://doi.org/10.1016/s2352-4642(22)00277-2)), some of which is brought in in the discussion section. It should be pulled into the introduction to set up the justification for why this study is a critical step in developing/adapting an ECD curriculum. In simple terms, statements like “Responsive caregiving and early learning activities positively impact early childhood development (ECD) outcomes, educational achievement and economic opportunities throughout the lifecourse [1–5]. Supporting caregivers to provide responsive care and learning activities during children’s early years is considered a key strategy for achieving equitable health and development outcomes.” Should be toned down just a bit, maybe by saying there seems to be strong evidence...”, and “a multi-country using DHS and MICS data indicates that... based on what is currently thought to be appropriate ECD services and supports”. Page 4 quote in 2nd paragraph: I think in Vancouver style direct quotes require the specific reference and page numbers. In this case it references 4 papers. And, as per the above point about being a bit more critical, do the authors agree with this statement? What does it actually mean in practice? If I read correctly, the authors somewhat contradict and/or expand on this statement in the discussion? This quote on its own without any critical framing confuses me and makes me cringe.

Methods: the methods section could benefit from a bit more structure and specificity. Here are the questions that came up for me in reading it: in-depth interviews, as in semi-structure in-depth? The FDGs were done separately with moms, fathers, and grandmothers, not mixed groups? How was ethnicity captured in the FDGs and interviews? The community member identified by village leaders is the person thereafter referred to as informant? Who were the supervisors? Were the supervisors the ones who did interviews? What determined a household to be “relevant”? Regarding the consent procedure, were all participants literate (enough) to do written consent? What were the ethnicities? Ethnic diversity is mentioned multiple times but there are no specifics presented.

In addition it would facilitate reading if the sampling approaches for SSIs and FDGs were separated and clearly differentiated. There is a mention that only participants over 18 were included. I am well aware of the implications and additional hoops to jump through when including under age participants. However, I believe BF has a fairly high rate of adolescent parenthood. Aside from the question of how to treat underage parents from a research ethics perspective, under age parents tend to be systematically excluded from research (and sometimes services and programs), resulting in a significant gap in our knowledge base. I would like to see justifications for this exclusion, and am sure there are many reasonable arguments.

Transcription and translation: pull up the mention of google translate from French to English; it comes late in the paragraph and the whole time I’m wondering how you did English analyses on French transcripts. Great though that you analyzed in both languages, with many checks and balances in place to ensure nothing got lost in translation from audio to transcript to translation. The last sentence of that transcription/translation/analysis paragraph would be better placed in the following paragraph.

	It would be nice to come back to the point about results not being shared with communities in the discussion. Could this be a potential limitation? Or an area for future work? Verifying results with the participants and/or their communities can bear great value. Discussion: The description of how this study influenced the intervention design is great. I think it could be made more prominent. Instead of several paragraphs summarizing the results followed by paragraphs describing how it influenced the intervention design, these things could be interwoven. And this is where you could come back to that statement about “blending” traditional with evidence based approaches, demonstrating that this study didn’t just allow you to identify “traditional approaches”, but actually provided entry points where what the ECD community thinks of as effective early stimulation programs align with, build on, and expand local perceptions and practices. (I don’t like the term “traditional” in this context, especially when juxtaposed with evidence-based, it’s patronizing)
--	--

VERSION 1 – AUTHOR RESPONSE

Reviewer 1

Dr. Khuthala Mabetha, University of the Witwatersrand Johannesburg

Abstract

Introduction

1. The introduction of the abstract has been well written but there are two important pieces of information missing: (1) From the whole background provided on responsive caregiving and its positive impact on developmental outcomes, what was the gap that merited further research into this area? i.e., gap related to why you conducted the study? (2) Secondly, what is the aim of the study? Please make the aim of the study explicit and place it right at the end of the introduction section.

Authors’ response: *Thank you for pointing out these gaps; we have added text in the Abstract Introduction (lines 57-59) to clearly state the gap our research addresses and we have moved the aim of the study from the methods section to the introduction section.*

Methods

2. “We conducted qualitative research in rural Burkina Faso to explore norms, beliefs, practices and aspirations”, lines 21-22: I see that the aim of the study was placed in the methods section. Please move it up right at the end of the introduction section.

Authors’ response: *See comment above.*

3. Also, what stood out to me is that the title indicates that the authors explored “perceptions” of young children’s learning and childhood development, yet the aim was to explore

“norms, beliefs, practices and aspirations”. All these terms are different to “perception” which refers to the manner in which a phenomenon is understood or interpreted whereas “beliefs” refers to the acceptance that something exists or is true. Same applies with “perception” and “practices” and “perceptions” and “aspirations”. The title is fine but the “perceptions” term needs to be replaced with a term that captures the aim of the study “norms, beliefs, practices and aspirations”.

Authors’ response: Thank you for raising the importance of this nuanced language; we have changed the title to be more aligned with the study aims: **‘Children awaken by playing’: A qualitative exploration of caregivers’ norms, beliefs and practices related to young children’s learning and early childhood development in rural Burkina Faso**

Main text

Introduction

1. “aged three four years”, line 3 – I think this was meant to be three to four years old. Please fix it.

Authors’ response: Thank you for spotting this typo. On the advice of Reviewer #2 we have significantly altered the Introduction and have subsequently removed this line.

2. “(Clinicaltrials.gov identifier NCT05335395”, line 28: This referencing style (if it is one) is different to the numbering system used throughout. Please be consistent (that’s if this is a referencing style and not a trial registration number). If it is a trial registration number, please be explicit and indicate that it is a trial registration number.

Authors’ response: This is indeed a trial registration number; we have made this explicit in the text (line 114).

Methods

1. “We conducted in-depth interviews (IDIs) with mothers and fathers and focus group discussions (FGDs)”, line 32- A few questions:
(a) How many IDIs and FGDs were conducted? How many mothers and how many fathers?

Authors’ response: Much of this information is provided further down in the section, in **Table 1**. To better direct the reader to this information, we have added (**see Table 1**) at the end of the section’s introductory sentence (line 121).

Given our use of the TIPS methodology with FGDs, we conducted two FGDs per participant group. We made this more explicit in the paragraph describing TIPS in detail (lines 153-154): “Two FGDs were conducted, in a neutral space away from other community members, with each participant group in each village by a facilitator and note taker.”

2. What was the maximum number of people in each FGD?

Authors' response: Thank you for pointing out this important piece of missing information. We have now incorporated the 5-7 participant FGD size into **Table 1**.

3. Did each FGD constitute only of mothers, fathers and only of grandmothers OR was there a combination of both mothers, fathers, and grandmothers? What was the rationale for this combination or lack thereof?

Authors' response: Your comment made us aware that this detail was not clear in the text; we have added text to clarify your questions above (from line 152):

"FGDs were stratified by participant group to allow topics to be discussed from a perspective of similar positionalities."

4. Given that IDIs were conducted with mothers and fathers of the children AND FGDs were also conducted with the mothers and fathers, including grandmothers, what was the rationale for using both data collection methods? Please elaborate on this.

Authors' response: We fully agree it would be beneficial for the reader to understand our justification for and use of the two different data collection methods. We have elaborated in lines 118-119:

"FGDs allowed us to explore social norms and beliefs and IDIs to understand how these played out in caregivers' lived experiences."

5. How was data saturation reached? In other words, did you get to nth number of in-depth interviews perhaps due to redundancy in information provided by the participants or was nth number the sample size selected for a specific reason? Same applies to the FGD discussions.

Authors' response: Thank you for raising this point, we realize that we were not as clear about our specific sampling process as we would like to be. We asked our research teams to meet a specific sample size for each data collection method ahead of field work. This sample size per data collection method was based on recommendations from existing evidence on adequate sample size (Guest et al, 2006) and our team's previous experience in and familiarity with the research context (Hollowell et al, 2019). The number of IDIs and FGDs we planned are the numbers included in Table 1. During data analysis we held frequent discussions to determine if we felt saturation had been reached (additional data collection would not have produced new information); about 75% of the way through analysis the team determined that saturation had in fact been reached and that additional IDIs and FGDs would not be necessary. We have made this more explicit in lines 201-204:

"We held frequent meetings to discuss findings, clarify translations, iteratively refine the code books and confirm that saturation was reached within our planned sample size (additional data collection would not have produced new information)."

6. “in four villages in Balé and Boulgou provinces, Burkina Faso”, lines 33-34: I would suggest that you construct a table showing how many people were sampled in each of these four villages for both data collection methods (also show how many were mothers, fathers, grandmothers) as a supplementary table to refer to. I see you have Table 1 showing the number of participants but I think you should also show how many were sampled in each of these of these four villages for both data collection methods.

Authors’ response: *Thank you for the suggestion to give our readers a more precise idea of our sampling by geography/ socio-cultural context. Given that we sampled an equal number of participants from each village, we felt it would make the in-text table unnecessarily crowded if we showed participants by village. Instead, we added the following specification at lines 119-121: “Data were collected in four villages in Balé and Boulgou provinces, with equal numbers of participants sampled from each village (see Table 1).”*

7. “The Burkina Faso Institute for Health Science Research Ethics Committee and the University College London Research Ethics Committee granted ethical approval for this study”, lines 42-43: Sub-heading missing. Please indicate what this refers to. Either “ethical considerations” or “ethics”.

Authors’ response: *Thank you for this recommendation to improve the organization of this section; we have now added the sub-heading ‘Ethics’ (line 125).*

8. “No participants left the study once they agreed to participate” line 13-14- I am not sure what the authors mean here. Are they saying once participants agreed to participate, they did not have the option to opt out of the study (suggesting that participation was only voluntary when consenting) or does this mean that out of all the participants who agreed to participate, none of them opted out of the study? Please correct this so that there is no ambiguity or more so that it is clear that participants were not forced to complete the study.

Authors’ response: *Thank you for highlighting that this line was not clear. Upon further consideration, we felt it was more relevant/ important for the reader to know how many people we lost to follow up. We have now added the following in lines 167-168: “Four participants were lost to follow up in the second FGDs (two fathers, one mother, and one grandmother).”*

9. “Data collectors were paired with participants of their same gender.”, lines 18-19- What was the reason for this?

Authors’ response: *We have made our justification for this decision more clear by adding the following (lines 171-172): “Given local gender power dynamics which might have induced response bias between researchers and participants of different genders, data collectors were paired with participants of their same gender.”*

10. Of importance, were the IDIs and FGDs conducted only in English or were there instances where participants were interviewed in their native language?

Authors’ response: *We noted that research activities were conducted in Bissa or Dioula but realize that it was not necessarily obvious that these were local languages. To make this clear, we have adjusted lines 170-171 to read: “Research activities lasted 45-90 minutes, were conducted in local languages (Bissa or Dioula) and audio recorded.”*

11. “deductive-inductive approach”, line 28- I understand the use of an inductive approach but I am not sure why a deductive approach was also employed since there was no mention of a predetermined framework that was used in this study which also contributed to the development of the IDIs and FGD topic guides. The authors only mentioned that this research informed the development of Scaling Up Nurturing Care, a Radio Intervention to Stimulate Early Childhood Development towards the end of the introduction but did not explain how it informed its development. Thus, a deductive approach is not appropriate in this context.
12. Please briefly explain the steps that you undertook in the inductive analysis which ultimately led into the generation of the themes and sub-themes.

Authors’ response: *We agree that we needed to expand on the description of our analysis approach as we did not use a framework to inform our deductive approach, but we were rather referring to the initial themes covered in our interview guides. We have made this more clear and expanded our description of our analysis process in lines 192-205: “Transcripts were analysed by francophone (MB, MD, SK, PO) and anglophone (ZH, MD) senior researchers. Non-francophone researchers translated transcripts using Google Translate. We used a deductive-inductive approach to content analysis. First, team members read three IDI and three FGD transcripts to ensure familiarization with data and themes. A bilingual researcher deductively developed separate initial code books for each data collection method based on the topics covered in the research guides. Then, a selection of IDIs and FGDs were double coded inductively within these broad themes to develop sub-themes and facilitate standardized coding between team members. The code books were elaborated via double coding a selection of transcripts. The team discussed any variation in coding and resolved differences via consensus discussion. Once coding was standardized and adequately detailed, the remaining transcripts were divided between researchers for analysis. We held frequent meetings to discuss findings, clarify translations, iteratively refine the code books and confirm that saturation was reached within our planned sample size (additional data collection would not have produced new information). Francophone researchers used NVIVO (released March 2020) to complete analysis; the anglophone researcher coded translated transcripts (via Google Translate) by hand.”*

Reviewer 2

Dr. Alice Wuermli , New York University

Comments to the Author:

This paper presents the process and findings of a qualitative study to inform the design and curriculum of a radio-based ECD intervention aimed at encouraging early stimulation. I very much appreciate this contribution and I thank the authors for the opportunity to provide my 5 cents. The paper is well written and structured, and methodologically as far as I can tell sound (with some requests for additional specificity and clarifications). Below I suggest areas for improvement.

1. The paper points out the cultural biases in the research on child development in the Majority World and proposes to address these in this study in order to inform a culturally adapted, appropriate version of a radio program to promote ECD. Overall, especially later on in the Discussion, the paper does a good job at demonstrating how this process informed the design of this intervention. Earlier on though there are a few instances where I would like to see a bit more critique or caution.

How are “we” defining and measuring responsive/sensitive caregiving and stimulation?
 How are “we” measuring child outcomes? Are these concepts and measures universally applicable and unbiased? And are we seeing impacts in interventions encouraging “stimulation” because stimulation always leads to better outcomes, or because our

outcome measures are picking up on what we are “teaching” in these interventions? Just because an intervention targeting a certain behavior leads to better outcomes according to our measures doesn’t mean that the relationship holds universally, or that the research is culturally appropriate and valid.

I would like to see the authors be a bit more critical in how they conceptualize development, developmental processes, and developmental outcomes and couch existing evidence within this framework. I get the sense that the authors understand and appreciate the cultural biases in ECD research, and I’d like to see this reflected throughout the paper. Eg., if outcomes are assessed with a Bayley, what does that mean for our ability to claim better child outcomes? Is the Bayley universally predictive of later outcomes? Who decided that the tasks on the Bayley are what matters, at a given age and stage of development?

Furthermore, the Lu et al. paper uses DHS and MICS data which despite their tremendous contribution have not gone without criticism of being somewhat culturally insensitive.

There is an extensive literature on culture and ECD that should be reflected (see for instance Scheidecker, G., Chaudhary, N., Oppong, S., Röttger-Rössler, B., & Keller, H. (2022). Different is not deficient: respecting diversity in early childhood development. *The Lancet Child & Adolescent Health*, 6(12), e24-e25. [https://doi.org/10.1016/s2352-4642\(22\)00277-2](https://doi.org/10.1016/s2352-4642(22)00277-2)), some of which is brought in in the discussion section.

It should be pulled into the introduction to set up the justification for why this study is a critical step in developing/adapting an ECD curriculum.

In simple terms, statements like “*Responsive caregiving and early learning activities positively impact early childhood development (ECD) outcomes, educational achievement and economic opportunities throughout the lifecourse [1–5]. Supporting caregivers to provide responsive care and learning activities during children’s early years is considered a key strategy for achieving equitable health and development outcomes.*” should be toned down just a bit, maybe by saying there seems to be strong evidence...”, and “a multi-country using DHS and MICS data indicates that... based on what is currently thought to be appropriate ECD services and supports”.

Authors’ response: *Thank you for these critical reflections and comments. Our original introduction to this paper was, in fact, a bit stronger in its critique of aspects of ECD programming and evaluation. We appreciated your recommendation to return to some of those original, provocative ideas. We have reworked the introduction guided by your useful comments and feedback including removing estimates of child ‘underdevelopment’ in the majority world which we agree are problematic, especially when taken out of context of a discussion as to how and by whom the analyses are made. We have also incorporated where and as appropriate the literature you suggested from the Lancet series on child development, thank you for pointing us to this important conversation. We look forward to your feedback on this rewritten section.*

2. Page 4 quote in 2nd paragraph: I think in Vancouver style direct quotes require the specific reference and page numbers. In this case it references 4 papers. And, as per the above point about being a bit more critical, do the authors agree with this statement? What does it actually mean in practice? If I read correctly, the authors somewhat contradict and/or expand on this statement in the discussion? This quote on its own without any critical framing confuses me and makes me cringe.

Authors’ response: *Upon further reflection and reworking the Introduction we have deleted this quote, but have also ensure other direct quotes are properly cited with page numbers. Those*

mistakes were formatting oversights from our citation software. Thank you for bringing this to our attention.

Methods

“The methods section could benefit from a bit more structure and specificity. Here are the questions that came up for me in reading it.”

1. In-depth interviews, as in semi-structure in-depth?

Authors’ response: *Yes, this specification was added in line 117.*

1. The FDGs were done separately with moms, fathers, and grandmothers, not mixed groups?

Authors’ response: *Yes, FGD were conducted with homogenous participant groups. The other reviewer had the same question, and we have made this clearer in lines 152-153.*

2. How was ethnicity captured in the FDGs and interviews? What were the ethnicities? Ethnic diversity is mentioned multiple times but there are no specifics presented.

Authors’ response: *Participants were asked to provide their ethnicity, if they felt comfortable, before the start of each interview along with other basic demographic information. We have specified the number of ethnic groups represented within the sample population in lines 180-181: “Ten ethnic groups were represented in the sample population, with about half of participants identifying as Bissa.” Due to our constrained word limit we did not elect to list all ethnicities individually but did specify, as written in lines 175-176, the ethnicity with which the highest percentage of participants identified.*

3. The community member identified by village leaders is the person thereafter referred to as informant?

Authors’ response: *Yes, to make this more clear we slightly edited line 137-142 to read: “Supervisors used the same procedure to recruit for IDIs and FDGs: First they introduced the study to village leaders who identified a community informant knowledgeable about the community, but not a member of the village chief’s family. The informant, often a community health worker, then accompanied the supervisor to households with eligible children, where supervisors introduced the study and provided an information sheet detailing the study objectives, confidentiality, participants’ rights and research team contact numbers.”*

4. Who were the supervisors? Were the supervisors the ones who did interviews?

Authors’ response: *To make supervisors’ roles more clear we have added details to the paragraph beginning at line 130 Data Collection:*

“Data were collected from March-April 2021. Data collectors and supervisors (women and men) had completed two to four years of undergraduate study and had previous experience collecting qualitative data. They underwent three weeks of training to review methods, ethics, research tools and transcription. We conducted a pilot study in Ouagadougou to finalise the data collection guides.

Supervisors were responsible for logistics, recruitment and data quality; data collectors obtained informed consent and conducted IDIs and FGDs.”

5. What determined a household to be “relevant”?

Authors’ response: *To make this point more clear we have edited lines 139-142 to read: “The informant, often a community health worker, then accompanied the supervisor to households with eligible children, where supervisors introduced the study and provided an information sheet detailing the study objectives, confidentiality, participants’ rights and research team contact numbers.”*

6. Regarding the consent procedure, were all participants literate (enough) to do written consent

Authors’ response: *We elaborated on the consent process in lines 142-144: “For participants unable to read French, information and consent forms were read aloud with a non-research team member witnessing and co-signing. For those unable to write, a thumb print replaced a signature.”*

7. In addition it would facilitate reading if the sampling approaches for SSIs and FDGs were separated and clearly differentiated.

Authors’ response: *We used the same sampling approach for both IDIs and FGDs, and due to word limitations we were not able to include separate explanations for each research activity; field supervisors were responsible for purposively selecting/ inviting participants to IDIs and FGDs to fulfill our target selection criteria for each research activity, as described in lines 150-154. We appreciate that this might not have been clear in our first submission, so have made this more clear for the reader by explicitly stating in lines 137: “Supervisors used the same procedure to recruit for IDIs and FGDs.”*

8. There is a mention that only participants over 18 were included. I am well aware of the implications and additional hoops to jump through when including under age participants. However, I believe BF has a fairly high rate of adolescent parenthood. Aside from the question of how to treat underage parents from a research ethics perspective, under age parents tend to be systematically excluded from research (and sometimes services and programs), resulting in a significant gap in our knowledge base. I would like to see justifications for this exclusion, and am sure there are many reasonable arguments.

Authors’ response: *Thank you for raising this point, and yes --- the logistics of speaking with individuals under the age of 18, considered by the ethics bodies giving approval for this study to be a ‘vulnerable’ population, were at the root of our decision to only speak with individuals (additional ethical approvals from both the UK and Burkina Faso on a tight timeline during ongoing interruptions due to the COVID-19 pandemic). We appreciate and agree with your points --- according to the 2021 DHS 11.7% of 17 year old Burkinabe women have had at least one pregnancy. We have raised this as an important limitation in lines 470-472 and appreciate you raising this point: “Logistical challenges also prevented us from speaking to individuals younger than 18 years old, an important limitation given high rates of adolescent pregnancy in Burkina Faso [52].”*

9. Transcription and translation: pull up the mention of google translate from French to English; it comes late in the paragraph and the whole time I'm wondering how you did English analyses on French transcripts. Great though that you analyzed in both languages, with many checks and balances in place to ensure nothing got lost in translation from audio to transcript to translation.
The last sentence of that transcription/translation/analysis paragraph would be better placed in the following paragraph.

Authors' response: *Thank you for raising this point, we have moved the specification of translation for the non-francophone researcher up in the paragraph and agree the last sentence of the paragraph belongs in the section on participant and public involvement (lines 192-210).*

Discussion

1. It would be nice to come back to the point about results not being shared with communities in the discussion. Could this be a potential limitation? Or an area for future work? Verifying results with the participants and/or their communities can bear great value.

Authors' response: *We fully agree, and have also added this point to our Limitations paragraph including a citation of an ECD co-design intervention (lines 475-477): "In future, the participation of local populations at every stage of formative research and intervention co-design [54] could be key in realizing the highest level of ethical, culturally adapted, effective ECD programming."*

2. The description of how this study influenced the intervention design is great. I think it could be made more prominent. Instead of several paragraphs summarizing the results followed by paragraphs describing how it influenced the intervention design, these things could be interwoven. And this is where you could come back to that statement about "blending" traditional with evidence based approaches, demonstrating that this study didn't just allow you to identify "traditional approaches", but actually provided entry points where what the ECD community thinks of as effective early stimulation programs align with, build on, and expand local perceptions and practices. (I don't like the term "traditional" in this context, especially when juxtaposed with evidence-based, it's patronizing)

Authors' response: *Thank you for these insightful comments and reflections. We have significantly edited the Discussion in response to this feedback, and look forward to your thoughts on what we see to be significant improvements to this final section of the paper. Given the significant changes made, we have highlighted the entire Discussion section for your re-review.*

VERSION 2 – REVIEW

REVIEWER	Mabetha , Khuthala University of the Witwatersrand Johannesburg
REVIEW RETURNED	03-Oct-2023
GENERAL COMMENTS	Under the "Ethics sub-heading", please add the ethics approval number.
REVIEWER	Wuermli , Alice New York University
REVIEW RETURNED	19-Sep-2023

GENERAL COMMENTS

I thank the authors for the careful consideration and thoughtful incorporation of reviewer feedback. Overall the changes have significantly strengthened the paper. It reads well, the methods section is significantly clearer, and the discussion section nicely presents how this research fits within a larger intervention development and adaptation endeavor. Below are some questions and suggestions that, in my opinion, would strengthen the paper even more.

Abstract:

Last sentence in Methods para in abstract should go in intro para. It's not a methods this study used.

Introduction:

Definitely improved! I'd like to push you a tiny bit further. I don't think this requires substantial rewriting or adding of content, rather it's about sharpening the arguments. The introduction now includes a significant number of references. But it feels a bit short and superficial. I'm missing the overarching why, the nuance in what you are arguing. For instance, what does it mean for an intervention to be ethical? What exactly are we worried about should an intervention not be properly adapted? What is our goal as researchers/interventionists? Regardless of where we are from and what cultural beliefs we hold, what is it we're trying to achieve? I'm playing devils advocate here. Do we really think a poorly adapted radio show would damage beliefs, values, and social cohesion? Maybe. But I might argue, more likely the messages on ECD would be dismissed, deemed irrelevant, and possibly strengthen social cohesion in response to just another attempt of neocolonial power grabbing and manipulation (people would turn off the radio). My sense is that we all here believe that talking to and playing with a baby from the very first days is beneficial. Then isn't the goal to suggest and ultimately convince people of the benefits, even if there are strong held norms that work against such behaviors currently? If so, what exactly is the point of understanding local norms etc. about ECD? The people believe the earlier the child starts to awaken, the better, and that caregivers can influence child awakening. How does this help us regarding adaptation of an intervention? I would like to encourage the authors to try to strengthen and sharpen the introduction in a way that reflects what it is they are trying to achieve, and is reflective of their own convictions that are motivating this work. Last sentence second paragraph: The paragraph is about interventions, and this last sentence is about research biases. Maybe the nuance is missing, but this doesn't seem to follow from the rest of the paragraph.

Line 118: Which countries exactly? And what was the outcome?

Methods:

Translation and transcription: audio was transcribed from local language directly to French? Or was the audio first transcribed in local language, then into French?

Did any of the Francophone researchers review the Google translations?

Line 244-245: shouldn't this go into the following paragraph?

Line 313: The second part of the sentence is grammatically off. I would insert a full-stop after "down", delete "and", use the plural "were".

Line 357 onward: this almost feels like its own section. FDG responses with regard to the TIPS activities

	Results: Overall the results could benefit from more structure. There are sections with guiding titles, but within them there is often a mix of data presented. For example, in the section “parents’ aspirations for their children”, the first couple of paragraphs and quotes are about aspirations. But then it goes into how parents can facilitate this, which is more along the lines of when and how children awaken and the “future impacts” of caregiver behaviors, and then “poverty” being a major obstacle for supporting their children. Right now the reader is left to their own devices to figure out the main takeaways. It reads a little like a running list of points. Discussion: Line 451-452: “Radio messages also integrate local beliefs which might hinder the practice of ECD behaviors.” How so? Which messages are these? Last sentence of that paragraph starting line 453: this belongs above to the other parts about how SUNRISE encourages early interactions. The explanation of “logistical challenges” that prevented the inclusion of underage mothers in the responses to the reviewer comments is more specific. I’d recommend using that explanation. Just logistical challenges doesn’t say much. Line 485: new paragraph for “A limitation...”. Same sentence, TIPS is suddenly with a small s (“TIPs”) 486: minor tweak to sentence to clarify what the social desirability bias refers to, e.g., “... behaviors, resulting in social desirability bias in participant responses.” Is there anything to demonstrate how your methods to prevent social desirability bias were effective? Line 488: what does “normalizing potential challenges” mean? 495-496: acknowledging and addressing / adapting to / other? I don’t think you mean to “integrate differences”? Equitable child development... across contexts? The idea is that if ECD programming is adapted to each and every context, taken together we’d have more equitable outcomes at a global level, not that adaptations in one context would lead to outcomes globally?
--	--